# A Comparison of Sanger Sequencing and Amplicon-Based Next Generation Sequencing Approaches for the Detection of HIV-1 Drug Resistance Mutations

**DOI:** 10.3390/v16091465

**Published:** 2024-09-14

**Authors:** Camilla Biba, Lia Fiaschi, Ilenia Varasi, Chiara Paletti, Niccolò Bartolini, Maurizio Zazzi, Ilaria Vicenti, Francesco Saladini

**Affiliations:** Department of Medical Biotechnologies, University of Siena, 53100 Siena, Italy; camilla.biba@student.unisi.it (C.B.); lia.fiaschi@unisi.it (L.F.); ilenia.varasi@student.unisi.it (I.V.); c.paletti@student.unisi.it (C.P.); niccolo.bartolini@student.unisi.it (N.B.); zazzi@unisi.it (M.Z.); saladini6@unisi.it (F.S.)

**Keywords:** HIV genotype, drug resistance mutations, NGS, antiretroviral

## Abstract

Background: Next-generation sequencing (NGS) kits are needed to finalise the transition from Sanger sequencing to NGS in HIV-1 genotypic drug resistance testing. Materials and Methods: We compared a homemade NGS amplicon-based protocol and the AD4SEQ HIV-1 Solution v2 (AD4SEQ) NGS kit from Arrow Diagnostics for identifying resistance-associated mutations (RAMs) above the 5% threshold in 28 plasma samples where Sanger sequencing previously detected at least one RAM. Results: The samples had a median 4.8 log [IQR 4.4–5.2] HIV-1 RNA copies/mL and were mostly subtype B (61%) and CRF02_AG (14%). Homemade NGS had a lower rate of samples with low-coverage regions (2/28) compared with AD4SEQ (13/28) (*p* < 0.001). Homemade NGS and AD4SEQ identified additional mutations with respect to Sanger sequencing in 13/28 and 9/28 samples, respectively. However, there were two and eight cases where mutations detected by Sanger sequencing were missed by homemade NGS and AD4SEQ-SmartVir, respectively. The discrepancies between NGS and Sanger sequencing resulted in a few minor differences in drug susceptibility interpretation, mostly for NNRTIs. Conclusions: Both the NGS systems identified additional mutations with respect to Sanger sequencing, and the agreement between them was fair. However, AD4SEQ should benefit from technical adjustments allowing higher sequence coverage.

## 1. Introduction

In recent years, advances in antiretroviral therapy (ART) have significantly improved the life expectancy for people living with HIV (PLWH). Indeed, several drugs targeting different steps of the HIV-1 replication cycle are now available, with improved convenience, reduced toxicity and higher genetic barrier, while several other investigational antiretrovirals have shown promising results in clinical trials [1]. Despite these improvements, all the currently available drugs, including those belonging to the newer classes, are at risk of losing their full efficacy due to emergence of resistance mutations, thus compromising the success of long-term therapy [2]. Indeed, real-world evidence shows that even the second-generation integrase inhibitor dolutegravir, which is recommended by the WHO as the preferred drug for first-line and second-line ART in all populations due to its high genetic barrier to resistance [3], can occasionally lead to the selection of resistance mutations within integrase, particularly in the presence of nucleoside reverse transcriptase inhibitor resistance [4]. For these reasons, the international guidelines recommend the execution of HIV-1 genotypic tests on viral RNA to evaluate transmitted and acquired drug resistance at HIV diagnosis and at treatment failure, respectively (EACS guidelines v12.0).

Dideoxynucleoside Sanger sequencing has been the standard method for HIV-1 drug resistance testing in clinical settings over the last two decades [5]. Despite widespread use and clinical effectiveness, Sanger sequencing identifies the predominant HIV-1 quasispecies but fails to detect mutations occurring at frequencies lower than 20–30% of the viral population [6]. This limitation has been overcome with the advent of next-generation sequencing (NGS). NGS can detect mutations that occur at frequencies lower than 5% with an accuracy greater than 99% [7]. NGS includes different applications that allow the sequencing of specific genomic fragments as well as of the whole virus genome [8]. Thus, most laboratories are transitioning from Sanger sequencing to NGS for HIV-1 genotypic drug resistance testing. While NGS platforms and chemistry are evolving at a fast pace, Illumina NGS technology, based on sequencing by synthesis, has gained widespread popularity due to low error rates [9] and high throughput with a relatively simple workflow at a reasonable cost [10]. However, until a few years ago, most NGS platforms were certified for research use only and thus were not suitable for clinical diagnostic applications. Recently, three NGS systems have been approved by the FDA for HIV-1 genotypic resistance testing in a diagnostic setting: the Sentosa^®^ SQ HIV Genotyping Assay by Vela Diagnostics (The Kendall, Singapore), the DeepChek^®^ Assay HIV-1 Full PR/RT/INT Drug Resistance system by ABL Diagnostics (Woippy, France) and the AD4SEQ HIV-1 Solution v2 by Arrow Diagnostics (Genoa, Italy). While the Sentosa^®^ assay is based on Ion Torrent technology (PMID: 35269868), both the assays by Arrow Diagnostics and ABL Diagnostics are based on Illumina technology. The aim of this study was to compare the sensitivity of the latter two NGS methods in identifying resistance-associated mutations (RAMs) detected by Sanger sequencing in clinical HIV-1 samples.

## 2. Materials and Methods

### 2.1. Samples

Access to residual anonymised plasma samples derived from clinical practice was initially obtained through patients’ informed consent as approved by the local Ethics Committee at the University Hospital of Siena. A total of 28 HIV-1 viremic plasma samples with a previous Sanger sequencing-based genotypic resistance test performed at the Microbiology and Virology Unity of the University Hospital of Siena, Italy, were selected. All samples had at least one RAM to protease inhibitors (PIs), nucleoside and non-nucleoside reverse transcriptase inhibitors (NRTIs and NNRTIs) or integrase inhibitors (INIs). The viral load of each sample was available as quantified by the Cobas^®^ 6800 system (Roche Diagnostic, Basel, Switzerland) for routine testing.

### 2.2. HIV-1 RNA Extraction and Amplification

Viral RNA was extracted from 1 mL of plasma with the eMAG automated system (bioMérieux, Craponne, France). HIV-1 RNA extracts were reverse-transcribed with the ImProm-II Reverse Transcriptase (Promega, Madison, WI, USA). Two consecutive PCR reactions (outer and inner PCR, Table 1) were performed to amplify the reverse transcriptase (RT, amino acids 1-400), the whole protease (PR) and the whole integrase (IN) coding regions. The outer PCR included 5 µL of cDNA, 10 µL of 5× Colourless GoTaq^®^ Flexi buffer (Promega, Madison, WI, USA), 2.5 mM MgCl_2_, 80 µM dNTPs, 3 pmol of primers (Table 1), 1 U of GoTaq^®^ Flexi DNA Polymerase (Promega, Madison, WI, USA) and sterile DNAse-/RNAse-free water to a final volume of 50 µL. The thermal profile included a denaturation step of 4 min at 95 °C followed by 5 cycles of 56 °C for 30 s, 68 °C for 2 min and 95 °C for 40 s, 25 cycles of 54 °C for 30 s, 68 °C for 2 min and 95 °C for 40 s and then two final steps at 54 °C for 1 min and 68 °C for 8 min. A total of 2 µL of the outer PCR mixture was used as the template for the nested PCR including 6 µL of 5× Green GoTaq^®^ Flexi Buffer, 2.5 mM MgCl_2_, 80 µM dNTPs, 3 pmol of primers (Table 1), 1 U of GoTaq^®^ Flexi DNA Polymerase and sterile DNAse-/RNAse-free water to a final volume of 30 µL. The thermal profile included a denaturation step of 4 min at 95 °C, followed by 5 cycles of 56 °C for 30 s, 68 °C for 2 min and 95 °C for 40 s, 30 cycles of 54 °C for 30 s, 68 °C for 2 min and 95 °C for 40 s and then two final steps at 54 °C for 1 min and 68 °C for 8 min. A total of 5 µL of the inner PCR was loaded onto a 1.5% Seakem agarose gel and was run at 6 V/cm for 50 min, and the presence of expected bands was checked under a transilluminator after gel staining with GelRed dye solution (Biotium, Fremont, CA, USA). The amplicons obtained were used both for Sanger sequencing and for the homemade amplicon-based NGS protocol. 

### 2.3. Sanger Sequencing

All sequencing reactions were performed on a 3500xL Dx Genetic Analyzer (Applied Biosystems, Waltham, MA, USA). PCR amplicons (PR-RT and IN) were diluted to a final concentration of 1–3 ng/µL; then, 10 µL was purified by adding 2 µL of ExoSAP-IT for PCR Product Clean-Up (Affymetrix, Santa Clara, CA, USA) at 37 °C for 15 min, followed by an inactivation step at 80 °C for 15 min. Each sequencing reaction included 3 µL of purified PCR product, 3.2 pmol of sequencing primer (Table 1), 2 µL of BigDye^®^ Terminator v1.1 Ready Reaction Mix (Life Technologies, Waltham, MA, USA), 1 µL of 5× Sequencing Buffer and sterile DNAse-/RNAse-free water, giving a final volume of 10 µL. The thermal cycler profile for this reaction included an initial denaturation step at 94° for 4 min, followed by 25 cycles at 50 °C for 1 min, 68 °C for 4 min, and 94 °C for 1 min. Sequencing reactions were treated with the X-Terminator^®^ Purification kit (Applied Biosystems, Waltham, MA, USA) in a 96-well plate as suggested by the manufacturer and loaded into a capillary electrophoresis sequencer. Chromatograms were assembled and edited with the DNAStar 7.1.0 SeqMan module to generate the FASTA files that were submitted to the Stanford online HIVdb system (https://hivdb.stanford.edu/; accessed on 1 August 2024) for the interpretation of RAMs. 

### 2.4. Homemade Amplicon-Based NGS

Amplicons (Table 1) were obtained as described above for Sanger sequencing, purified with the Agencourt^®^ AMPure^®^ magnetic beads (Beckman Coulter, Brea, CA, USA), quantified with the Qubit fluorometer in dsDNA HS Buffer (Qiagen, Venlo, The Netherlands) and diluted to a final concentration of 0.8 ng/µL. PR-RT and IN amplicons derived from the same sample were pooled together for the subsequent steps. Tagmentation was performed with the Nextera^®^ XT DNA Library Preparation Kit (Illumina, San Diego, CA, USA), according to the manufacturer’s instructions. Tagmented samples were indexed with IDT for Illumina DNA/RNA UD Indexes (Illumina, San Diego, CA, USA) and pooled together to obtain the library that was diluted to a final concentration of 9 pM. The library was then spiked-in with 10% 9 pM PhiX control library (Illumina, San Diego, CA, USA), loaded on a Nano MiSeq Reagent Kit v2 (2 × 250 bp paired-end reads, 8.5 Gb output, Illumina, San Diego, CA, USA) and run on an Illumina MiSeq instrument.

### 2.5. AD4SEQ HIV-1 Solution v2 Kit

The RNA extracts were obtained as described above and, according to the AD4SEQ HIV-1 Solution v2 protocol (Arrow Diagnostic, Genoa, Italy), samples with viremia above 20,000 HIV-1 RNA copies/mL were five-fold diluted in sterile DNase-/RNase-free water. Then, one-step RNA reverse transcription was performed, followed by amplification of the target regions in two separate PCRs, defined as Target PCR. Target PCR amplifies the following regions of the HIV-1 genome: PR (codons 1–99), RT (codons 1–440), IN (codons 1–289) and gp120 (codons 266–366). Once the reaction was completed, amplification products were visualised by electrophoresis. Then, for each sample, the two separated Target PCR reactions were combined in a single tube and purified with magnetic beads (Agentcourt^®^ AMPure^®^ magnetic beads, Beckman Coulter, Brea, CA, USA). Individual samples were indexed using the index adaptors provided by the kit (Index PCR), purified as described above, and amplification was checked again by electrophoresis. The concentration of each purified Index PCR was determined with the Qubit fluorometer in dsDNA HS Buffer (Qiagen, Venlo, The Netherlands), and the molarity of each sample was calculated using the following formula: sample concentration [nM] = (sample concentration [ng/µL] × 10^6^)/(656.6 × AMPpb), where AMPbp is the average length of the amplicons in base pairs. After quantification, each sample was diluted to 20 pM, and 3 µL was pooled in the library. After purification, the library was diluted to a final concentration of 9 pM and spiked-in with 20% of 9 pM PhiX control library (Illumina, San Diego, CA, USA), loaded on a Nano MiSeq Reagent Kit v2 (2 × 250 bp paired-end reads, 8.5 Gb output, Illumina, San Diego, CA, USA) and run on an Illumina MiSeq instrument.

### 2.6. HIV-1 Subtyping and Drug Resistance Interpretation 

The assignment of HIV-1 subtypes was performed by the COMET HIV-1 tool [11]. NGS data generated by AD4SEQ were analysed both with the dedicated SmartVir software version 1.0.6 provided by Arrow Diagnostics (AD4SEQ-Smartvir) and by the online HIVdb Drug Resistance Database system (AD4SEQ-HIVdb) (Stanford University; https://hivdb.stanford.edu/; accessed on 1 August 2024). NGS data generated by the homemade system were analysed only by HIVdb. The FASTA files obtained from Sanger sequencing were analysed directly, while the FASTQ files obtained from NGS were first converted to CodFreq files and then analysed to detect RAMs. The CodFreq files list the frequency of the mutations of each nucleotide triplet within the sequenced region. For the analysis of FASTQ files, the minimum read depth parameter was set at >100 reads per position with a mutation detection threshold of 5%, both for SmartVir and for HIVdb, independently of whether the position was associated with drug resistance or not. The Stanford HIVdb 9.6 algorithm was used to infer drug susceptibility from the RAMs identified by the two NGS systems and by Sanger sequencing. Mutations were defined as RAMs when associated with a score for at least one drug or included in a combination rule changing the score for at least one drug in the HIVdb 9.6 algorithm. Agreement among the RAMs identified by the different systems was qualitatively defined when identical RAMs were identified. 

### 2.7. Statistical Analysis

Continuous variables were reported as median and interquartile ranges (IQRs). The difference in proportions was analysed by a chi-square test followed by a post hoc z test with the Bonferroni adjustment for multiple comparisons. The difference between numerical values obtained with paired samples were analysed by a Wilcoxon signed-rank test. All the analyses were performed using the SPSS version 20 package (IBM Corp., Armonk, NY, USA).

## 3. Results

### 3.1. Samples Included in This Study

All the 28 plasma samples included in this study (Table 2) had detectable viremia with a median viral load of 4.8 log [IQR 4.4–5.2] HIV-1 RNA copies/mL. Subtype B was identified in 17 (61%) cases and CRF02_AG in 4 (14%). 

### 3.2. Comparison between Homemade NGS and AD4SEQ: Identification of Drug Resistance Mutations

The RAMs identified by the three different sequencing methods, i.e., Sanger sequencing as well as AD4SEQ (data processing by SmartVir) and homemade (data processing by HIVdb) NGS, are listed in Appendix A. RAMs to PIs, NRTIs, NNRTIs and INIs were detected in 7, 19, 22 and 7 samples, respectively, by at least one method (Table 3). Homemade NGS and AD4SEQ-SmartVir identified additional mutations with respect to Sanger sequencing in 13/28 and 9/28 samples, respectively.

Agreement between the NGS methods and Sanger sequencing for PIs was observed in 24/28 cases (85.7%). In two cases (samples 5826 and 6570), both NGS methods detected additional RAMs with respect to Sanger sequencing. In two samples (5979 and 6813), homemade NGS was more sensitive in detecting additional RAMs with respect to Sanger sequencing and AD4SEQ-SmartVir. 

With NRTIs, complete agreement among the NGS methods and Sanger sequencing was observed in only 18/28 (64.3%) cases. In three cases (samples 5826, 5974 and 7312), homemade NGS detected additional mutations with respect to AD4SEQ-SmartVir and Sanger sequencing, while in one case (sample 7312), AD4SEQ-SmartVir detected an additional mutation with respect to homemade NGS and Sanger sequencing. However, in five cases (samples 5979, 6003, 6006, 6436 and 6493), Sanger sequencing and homemade NGS gave comparable results, and AD4SEQ-SmartVir failed to recognise some drug resistance mutations. Differently from Sanger sequencing and AD4SEQ-SmartVir, homemade NGS failed to identify mutations D67N and K219E in sample 156471. Agreement between the NGS methods with respect to Sanger sequencing was observed only in one case (sample 6835). A strong discordance among the three methods was observed in sample 7312. Indeed, Sanger sequencing detected the D67N, T69G and K219Q mutations, homemade NGS detected the additional mutation S68G and a deletion at position 69, while AD4SEQ-SmartVir detected a combination of mutations at position D67(N/E) and the T69NG mutation.

With NNRTIs, similar to NRTIs, 19/28 (67.9%) cases showed complete agreement across the NGS methods and Sanger sequencing. In four cases (samples 155974, 156436, 156493 and 156817), homemade NGS detected additional mutations with respect to AD4SEQ-SmartVir and Sanger sequencing, while in two cases (samples 6669 and 6813), AD4SEQ-SmartVir detected additional mutations with respect to homemade NGS and Sanger sequencing. Additional mutations detected by both NGS methods but not by Sanger sequencing were observed in three cases (samples 6592, 7312 and 7347).

For INIs, complete agreement among all methods was observed in 25/28 cases (89.3%). In one case (sample 6669), both NGS methods were in agreement and detected the minority RAM E157Q with respect to Sanger sequencing. Homemade NGS was more sensitive in detecting the additional minority RAM T97A with respect to Sanger sequencing and AD4SEQ-SmartVir in sample 7347, but AD4SEQ-SmartVir was more sensitive than homemade NGS with sample 6835, detecting L74M. 

In conclusion, AD4SEQ-SmartVir did not identify one or more RAMs detected by at least one of the other two systems in 2 (7.1%), 8 (28.5%), 4 (14.3%) and 1 (3.6%) samples for PIs, NRTIs, NNRTIs and INIs, respectively, while homemade NGS did not identify one or more RAMs in 0, 2 (7.1%), 2 (7.1%) and 1 (3.6%) samples detected by at least one of the other two systems for PIs, NRTIs, NNRTIs and INIs, respectively. Differently from HIVdb, AD4SEQ-SmartVir did not report the mutation S68G and failed to detect mutations identified by Sanger sequencing and homemade NGS with a frequency >20% in three cases. By contrast, homemade NGS-HIVdb gave a slightly different identification of RAMs at two amino acid positions that were detected as predominant by Sanger sequencing or with frequency >20% by AD4SEQ-SmartVir in sample 7312. 

### 3.3. Comparison of SmartVir and HIVdb NGS Data Processing

Notably, SmartVir and HIVdb generated different outputs for read depth, i.e., median read depth vs. max. and min. coverage, respectively (Appendix A). Other parameters, such as the sequenced regions with a coverage depth of <100 reads per base, were instead reported by both interpretation systems (Table 4). To compare the two NGS data processing methods, we analysed the FASTQ files obtained by AD4SEQ both with Smartvir (AD4SEQ-Smartvir) and with HIVdb (AD4SEQ-HIVdb). Homemade NGS generated low-coverage data in 2/28 samples, both subtype B, in the initial part of the integrase encoding region (Table 4). A coverage depth <100 was more frequent both with AD4SEQ-HIVdb (17/28 cases) and with AD4SEQ-SmartVir (13/28) with respect to homemade NGS (2/28) (*p* < 0.001) while the difference between AD4SEQ-HIVdb and AD4SEQ-SmartVir was not statistically significant. The median read depth obtained with homemade NGS and with AD4SEQ-HIVdb was comparable (2189 [IQR 1842–7809] reads vs. 4634 [3056–6427] reads, *p* = 0.151). With AD4SEQ-SmartVir, the lower coverage affected seven B and six non-B subtypes, mainly at RT codons 14–49 and 260–319 and IN codons 1–75 and 201–288. 

Disagreement between RAMs detected by SmartVir and HIVdb when processing AD4SEQ FastQ files was observed in 10 samples (35.7%; Table 5). HIVdb detected one additional PI RAM in one case (3.6%). For NRTIs, 1 mutation (3.6%) was detected by SmartVir and not by HIVdb, 8 mutations (28.6%) were detected by HIVdb and not by Smartvir, including S68G in 6/8 cases. For NNRTIs, two mutations (10.7%) at position V179 were detected by HIVdb but not by SmartVir. 

### 3.4. Prediction of Drug Susceptibility

Table 6 shows the impact of the RAMs not detected uniformly by the four methods (Sanger sequencing, homemade NGS, AD4SEQ-Smartvir and AD4SEQ-HIVdb) on the prediction of drug susceptibility. Globally, drug susceptibility predictions were fully concordant in 20/28 (71.4%) samples for all the drugs considered, including PIs (Atazanavir, ATV; Lopinavir, LPV; and Darunavir, DRV), NRTIs (Abacavir, ABC; Tenofovir Alafenamide, TAF; and Lamivudine/Emtricitabine, 3TC/FTC), NNRTIs (Doravirine, DOR; Rilpivirine, RPV; Etravirine, ETR; Efavirenz, EFV; and Nevirapine, NVP) and INIs (Dolutegravir, DTG; Bictegravir, BIC; and Cabotegravir, CAB). 

Homemade NGS gave discordant results with respect to AD4SEQ-SmartVir in three samples for NRTIs (5979, 6471 and 7312) and in one sample each for NNRTIs (1669) and for INIs (6835). AD4SEQ-SmartVir and AD4SEQ-HIVdb gave discordant results in one sample each for PI (ATV and LPV for 6835) and for NRTIs (ABC, TAF and 3TC/FTC for 7312). 

Considering Sanger sequencing as a reference, homemade NGS was in agreement with Sanger sequencing for susceptibility predictions against all INIs and PIs but gave discordant predictions in 4/84 (4.7%) cases for NRTIs (6471 and 7312 for one drug and 6835 for two drugs) and in 14/140 (10.0%) cases for NNRTIs (6592 and 7312 for five drugs, 6493 for three drugs and 7347 for one drug). AD4SEQ-Smartvir was discordant with Sanger sequencing in 5/84 (6.0%) cases for NRTIs (5979 for two drugs and 7312 for three drugs), in 16/140 (11.4%) cases for NNRTIs (6592 and 7312 for five drugs, 6493 for three drugs, 6669 for two drugs and 7347 for one drug) and in 1/84 (1.2%) cases for INIs (6835). AD4SEQ-HIVdb gave discordant susceptibility predictions in 1/84 (1.2%) cases for PIs and INIs (6835 in both cases), in 2/84 (2.4%) cases for NRTIs (5979 for two drugs) and in 16/140 (11.4%) samples for NNRTIs (6592 and 7312 for five drugs, 6493 for three drugs, 6669 for two drugs and 7347 for one drug). Notably, most of the discrepancies between either NGS method and Sanger sequencing were explained by a lack of detection of minority RAMs (<20%) by Sanger sequencing, namely, in 11 (39%), 7 (25%) and 9 (32%) out of 28 cases with homemade NGS, AD4SEQ-Smartvir and AD4SEQ-HIVdb, respectively.

## 4. Discussion

NGS has redefined genome sequencing techniques, and HIV-1 genotyping has been an early NGS application in the field of infectious diseases with the aim of identifying minority RAMs that cannot be detected by Sanger sequencing [12]. Since NGS is more diversified and complex than Sanger sequencing, standardisation plays a key role in the transition from Sanger sequencing to NGS in clinical settings. Indeed, recent progress in HIV-1 NGS has led to the CE-IVD certification of some commercial systems, including the Sentosa^®^ SQ HIV Genotyping Assay developed by Vela Diagnostics, the AD4SEQ HIV-1 Solution v2 developed by Arrow Diagnostics and the DeepChek^®^ Assay HIV-1 Full PR/RT/INT Drug Resistance developed by ABL Diagnostics. 

Due to the limits of the Sentosa^®^ SQ HIV kit—based on the Ion Torrent platform that is more error prone and expensive than the small footprint Illumina platforms [13]—and the slow introduction of the DeepChek^®^ HIV Assay in Italy, the AD4SEQ HIV-1 Solution v2 is currently the most widely used in Italy. However, to our knowledge, there are no published studies evaluating the AD4SEQ HIV-1 Solution v2 kit in comparison with other NGS-based HIV-1 genotyping techniques or with the reference Sanger sequencing. In this work, we compared Sanger sequencing, AD4SEQ and a homemade amplicon-based NGS protocol to analyse a panel of 28 plasma samples derived from routine analysis. For both NGS methods, we used an Illumina MiSeq platform. The NGS reads generated by the homemade and AD4SEQ systems were processed by the Stanford HIVdb online tool for sequence reads and by the SmartVir system bundled with AD4SEQ, respectively. In addition, AD4SEQ NGS reads were also processed by HIVdb to compare the output of HIVdb and SmartVir on the same raw data (AD4SEQ-HIVdb vs. AD4SEQ-SmartVir). For both the NGS systems and Sanger sequencing, the RAMs detected were interpreted by the HIVdb 9.6 algorithm.

Overall, the Sanger sequencing and NGS results were more concordant for PI and INI than for NRTI and NNRTI RAMs. A higher proportion of additional PI and NRTI RAMs was detected by homemade NGS compared with AD4SEQ-SmartVir. Notably, in 17.8% of the samples, AD4SEQ-Smartvir failed to detect resistance mutations against NRTIs which were also detected by Sanger sequencing. The most frequently missed mutation was S68G in the RT coding region. This mutation was also detected when the AD4SEQ data were analysed with the Stanford HIVdb system, indicating that some settings in the SmartVir pipeline likely led to the exclusion of this polymorphic mutation. Indeed, comparing the two NGS data processing systems on AD4SEQ sequences, AD4SEQ-HIVdb and AD4SEQ-SmartVir missed 3.6% vs. 15.5% of the RAMs detected by the other system, respectively. However, the lower rate of RAM detection by AD4SEQ cannot be attributed solely to the interpretation system. Indeed, a lower coverage with respect to homemade NGS was observed for AD4SEQ (samples with a coverage < 100 reads were 2, 13 and 17 for homemade NGS, AD4SEQ-SmartVir and AD4SEQ-Stanford, respectively). While we were not able to identify any factor associated with low coverage in our dataset, a recently published study based on a larger number of sequence data generated by the AD4SEQ kit showed that non-B subtype and low viremia are associated with low coverage [14]. Nevertheless, predictions of drug susceptibility were affected only for NRTIs and prevalently when AD4SEQ data were processed by SmartVir (4.7%, 6.0%, 2.4% of discordant prediction for homemade NGS, AD4SEQ-SmartVir, AD4SEQ-Stanford, respectively, with respect to Sanger sequencing). Reassuringly, we did not observe any large shift in the prediction of drug susceptibility from a high score to a low score (i.e., from sensitive to resistant or vice versa).

Several published works [12,15,16,17] have evaluated the performance of homemade NGS systems with respect to Sanger sequencing and obtained results similar to those shown in this study, consistent with the inherently higher sensitivity of NGS in detecting minority RAMs. While participation in external quality assessment programs is a recognised approach for validating homemade NGS methodologies, CE-IVD-approved kits are essential for clinical use. To date, only the Sentosa^®^ SQ HIV kit has been compared with the reference Sanger sequencing [17,18]. Thus, this is the first paper analysing the performance of AD4SEQ HIV-1 Solution v2 kit by Arrow Diagnostics. Although we had the possibility of selecting samples shown to harbour drug resistant viruses by Sanger sequencing, the number of samples was limited and did not allow a wide representation of HIV-1 subtypes. In addition, all but one sample had >1000 HIV-1 RNA copies/mL; thus, we could not test the threshold of sensitivity for the AD4SEQ amplification method. For these reasons, further validation experiments on a larger and more heterogeneous sample panel are advisable for completing the assessment of the system. Notwithstanding such limitations, this study conveys relevant information on the performance and caveats of the AD4SEQ system in clinical settings. 

## Figures and Tables

**Table 1 viruses-16-01465-t001:** Primers used to amplify the protease, reverse transcriptase (amino acid 1–400) and integrase coding regions for Sanger sequencing and homemade NGS. _F, forward primer; _R, reverse primer; PR, protease; RT, reverse transcriptase; IN, integrase.

Primer	Sequence (5′-3′)	Target Region	HIV-1 HXB2 Coordinates	PCR Round
P534_F	AAAARGGYTGTTGGAAATGTGG	PR-RT	2018–2039	Outer
P1069_R	TCCCAYTCAGGAATCCAGGT		3774–3793	
P688_F	CATGGGTACCAGCACACAAAGG	IN	4150–4171	Outer
P689_R	CCCAAATGCCAGTCTCTTTCTCCTG		5261–5285	
P535_F	GARAGRCAGGCTAATTTTTTAGGGA	PR-RT	2071–2095	Inner
P1070_R	AATCCAGGTRGCYTGCCAATA		3762–3782	
P690_F	AGGRATTGGAGGAAATGAACA	IN	4169–4189	Inner
P691_R	GGGATGTGTACTTCTGAACTTA		5192–5213	

**Table 2 viruses-16-01465-t002:** Subtype and viral load of the 28 plasma samples included this study.

Sample	Subtype	HIV-1 RNA log10 Copies/mL
5826	B	4.3
5974	F1	4.9
5979	B	2.1
6003	CRF02_AG	3.1
6006	F1	6.0
6084	B	4.9
6092	B	4.9
6107	B	4.6
6216	B	4.8
6222	B	6.3
6322	A6	5.4
6363	B	4.8
6408	B	4.9
6436	CRF02_AG	4.8
6471	B	3.7
6493	CRF01_AE	5.1
6570	B	4.1
6592	C	4.5
6669	CRF02_AG	5.0
6695	B	4.7
6750	B	5.3
6762	B	3.1
6813	B	5.0
6817	B	5.4
6835	B	4.3
6880	F1	6.3
7312	G	4.8
7347	CRF02_AG	5.9

**Table 3 viruses-16-01465-t003:** Drug resistance mutations to (A) PIs (protease inhibitors), (B) NRTIs (nucleoside reverse transcriptase inhibitors) and NNRTI (non-NRTI) and (C) INI (integrase inhibitors) identified. The table lists the samples in which RAMs were detected by at least one of the three sequencing methods, highlighted in bold. The frequency of each mutation identified by NGS is shown in brackets.

(A)			
Sample	Sequencing System and Data Processing Method	PIs	
**5826**	Sanger-HIVdb	V32I, L33F, M46I, I47V, I54M, Q58E, T74TP	
	Homemade NGS-HIVdb	V32I (99%), L33F (99%), M46I (99%), I47V (99%), **I50V (29%)**, I54M (99%), Q58E (99%), T74P (27%)	
	AD4SEQ-SmartVir	V32I (99%), L33F (99%), M46I (99%), I47V (99%), **I50V (24%)**, I54M (99%), Q58E (99%), T74P (45%)	
**5979**	Sanger-HIVdb	L33F, I84V	
	Homemade NGS-HIVdb	L33F (98%), **I54T (10%)**, I84V (98%)	
	AD4SEQ-SmartVir	L33F (98%), I84V (98%)	
**6084**	Sanger-HIVdb	L10F, M46I, T74P	
	Homemade NGS-HIVdb	L10F (98%), M46I (99%), T74P (64%)	
	AD4SEQ-SmartVir	L10F (98%), M46I (98%), T74P (69%)	
**6092**	Sanger-HIVdb	L90M	
	Homemade NGS-HIVdb	L90M (98%)	
	AD4SEQ-SmartVir	L90M (99%)	
**6408**	Sanger-HIVdb	L33F, M46L, L90M	
	Homemade NGS-HIVdb	L33F (99%), M46L (99%), L90M (98%)	
	AD4SEQ-SmartVir	L33F (98%), M46L (97%), L90M (99%)	
**6570**	Sanger-HIVdb	None	
	Homemade NGS-HIVdb	**K20T (14%)**	
	AD4SEQ-SmartVir	**K20T (9.6%)**	
**6813**	Sanger-HIVdb	Q58E	
	Homemade NGS-HIVdb	Q58E (99%), **G73S (7.3%)**	
	AD4SEQ-SmartVir	Q58E (98%)	
(B)			
**Sample**	**Sequencing System and Data Processing Method**	**NRTIs**	**NNRTIs**
**5826**	Sanger-HIVdb	K70T, M184V	None
	Homemade NGS-HIVdb	K70T (78%), **V75I (6.5%)**, M184V (96%)	None
	AD4SEQ-SmartVir	K70T (70%), M184V (99%)	None
**5974**	Sanger-HIVdb	**S68G**	K103N, N348I
	Homemade NGS-HIVdb	**S68G (99%), K219Q (7.1%)**	K103N (99%), **V106I (10%)**, N348I (99%)
	AD4SEQ-SmartVir	None	K103N (99%), N348I (99%)
**5979**	Sanger-HIVdb	**M41L**, D67N, T215Y	K103N, Y181I
	Homemade NGS-HIVdb	**M41L (99%)**, D67N (95%), T215Y (91%)	K103N (97%), Y181I (97%)
	AD4SEQ-SmartVir	D67N (98%), T215Y (88%)	K103N (99%), Y181I (90%)
**6003**	Sanger-HIVdb	**S68G**, M184V	K103N, K238N
	Homemade NGS-HIVdb	**S68G (99%)**, M184V (99%)	K103N (98%), K238N (99%)
	AD4SEQ-SmartVir	M184V (98%)	K103N (95%), K238N (96%)
**6006**	Sanger-HIVdb	**S68G**	K103N, N348I
	Homemade NGS-HIVdb	**S68G (99%)**	K103N (98%), N348I (98%)
	AD4SEQ-SmartVir	None	K103N (99%), N348I (99%)
**6084**	Sanger-HIVdb	D67N, T215C	None
	Homemade NGS-HIVdb	D67N (92%), T215C (97%)	None
	AD4SEQ-SmartVir	D67N (98%), T215C (97%)	None
**6092**	Sanger-HIVdb	M41ML, M184V, T215TNSY	None
	Homemade NGS-HIVdb	M41L (28%), M184V (97%), T215Y (23%)	None
	AD4SEQ-SmartVir	M41L (36%), M184V (97%), T215Y (13%)	None
**6107**	Sanger-HIVdb	K219N	Y181C
	Homemade NGS-HIVdb	K219N (99%)	Y181C (98%)
	AD4SEQ-SmartVir	K219N (98%)	Y181C (99%)
**6216**	Sanger-HIVdb	M41L, T215D	None
	Homemade NGS-HIVdb	M41L (97%), T215D (98%)	None
	AD4SEQ-SmartVir	M41L (94%), T215D (90%)	None
**6222**	Sanger-HIVdb	None	V106I, G190A
	Homemade NGS-HIVdb	None	V106I (91%), G190A (99%)
	AD4SEQ-SmartVir	None	V106I (92%), G190A (98%)
**6322**	Sanger-HIVdb	None	E138A, G190GS
	Homemade NGS-HIVdb	None	E138A (90%), G190S (41%)
	AD4SEQ-SmartVir	None	E138A (89%), G190S (38%)
**6363**	Sanger-HIVdb	M184V	V106A, F227L
	Homemade NGS-HIVdb	M184V (92%)	V106A (97%), F227L (93%)
	AD4SEQ-SmartVir	M184V (98%)	V106A (97%), F227L (97%)
**6408**	Sanger-HIVdb	T215V	None
	Homemade NGS-HIVdb	T215V (95%)	None
	AD4SEQ-SmartVir	T215V (98%)	None
**6436**	Sanger-HIVdb	**S68G**	K103N
	Homemade NGS-HIVdb	**S68G (85%)**	K103N (98%), **K238T (7.8%), N348I (6.8%)**
	AD4SEQ-SmartVir	None	K103N (99%)
**6471**	Sanger-HIVdb	M41L, **D67N**, M184V, L210W, T215Y, **K219KE**	L100I, K103N, N348I
	Homemade NGS-HIVdb	M41L (98%), M184V (98%), L210W (98%), T215Y (95%)	L100I (96%), K103N (98%), N348I (98%)
	AD4SEQ-SmartVir	M41L (98%), **D67N (19%)**, M184V (91%), L210W (98%), T215Y (92%), **K219E (9%)**	L100I (99%), K103N (97%), N348I (99%)
**6493**	Sanger-HIVdb	D67G, **S68G**, K70R, M184V, **T215I**, K219E	K103N, V108I, K238T, N348I
	Homemade NGS-HIVdb	D67G (99%), **S68G (97%)**, K70R (98%), M184V (98%), **T215I (87%)**, K219E (78%)	K103N (98%), V108I (98%), **V179D (7%)**, **Y181C (34%)**, K238T (98%), N348I (98%)
	AD4SEQ-SmartVir	D67G (98%), K70R (99%), M184V (92%), K219E (87%)	K103N (98%), V108I (99%), **Y181C (26%)**, K238T (97%), N348I (99%)
**6570**	Sanger-HIVdb	M41L, M184MV	None
	Homemade NGS-HIVdb	M41L (44%), M184V (43%)	None
	AD4SEQ-SmartVir	M41L (66%), M184V (70%)	None
**6592**	Sanger-HIVdb	None	E138EK
	Homemade NGS-HIVdb	None	**K101E (40%)**, E138K (23%)
	AD4SEQ-SmartVir	None	**K101E (13%)**, E138K (29%)
**6669**	Sanger-HIVdb	None	K103KN, V106M
	Homemade NGS-HIVdb	None	K103N (75%), V106M (24%)
	AD4SEQ-SmartVir	None	K103N (45%), V106M (54%), **Y181C (7%)**
**6695**	Sanger-HIVdb	None	A98G
	Homemade NGS-HIVdb	None	A98G (99%)
	AD4SEQ-SmartVir	None	A98G (90%)
**6750**	Sanger-HIVdb	V75M	E138A
	Homemade NGS-HIVdb	V75M (96%)	E138A (99%)
	AD4SEQ-SmartVir	V75M (95%)	E138A (97%)
**6762**	Sanger-HIVdb	D67N, K219Q	K103N, V179T, Y181C, H221Y
	Homemade NGS-HIVdb	D67N (90%), K219Q (98%)	K103N (99%), V179T (T 76%), Y181C (98%), H221Y (99%)
	AD4SEQ-SmartVir	D67N (92%), K219Q (98%)	K103N (99%), V179T (T 76%), Y181C (98%), H221Y (98%)
**6813**	Sanger-HIVdb	None	E138A, G190A, M230L
	Homemade NGS-HIVdb	None	E138A (99%), G190A (99%), M230L (98%)
	AD4SEQ-SmartVir	None	**K103N (7%)**, E138A (92%), G190A (95%), M230L (96%)
**6817**	Sanger-HIVdb	None	K103KN, **V179T**
	Homemade NGS-HIVdb	None	K103N (67%), **V106I (5.4%)**, **V179T (99%)**
	AD4SEQ-SmartVir	None	K103N (69%)
**6835**	Sanger-HIVdb	L210W, T215S	None
	Homemade NGS-HIVdb	**M41L (30%)**, L210W (99%), **T215DS (D: 19%, S 79%)**	None
	AD4SEQ-SmartVir	**M41L (19%)**, L210W (98%), **T215DS (D: 25%, S: 72%)**	None
**6880**	Sanger-HIVdb	None	E138G
	Homemade NGS-HIVdb	None	E138G (96%)
	AD4SEQ-SmartVir	None	E138G (97%)
**7312**	Sanger-HIVdb	**D67Δ**, T69G, K219Q	A98G, V106I
	Homemade NGS-HIVdb	**D67ΔN (Δ: 8.9%, N: 53%), S68G (52%), T69Δ (93%)**, K219Q (96%)	A98G (99%), V106I (94%), **Y181C (55%)**
	AD4SEQ-SmartVir	**D67NE (N: 8%, E: 88%), T69NG (N: 6%, G: 89%)**, K219Q (96%)	A98G (97%), V106I (98%), **Y181C (66%)**
**7347**	Sanger-HIVdb	None	K101E, Y181C, N348I
	Homemade NGS-HIVdb	None	K101E (99%), **V106I (16%)**, Y181C (99%), N348I (99%)
	AD4SEQ-SmartVir	None	K101E (99%), **V106I (15%)**, Y181C (96%), N348I (96%)
(C)			
**Sample**	**Sequencing System and Data Processing Method**	**INIs**	
**5979**	Sanger-HIVdb	T97A, E138K, G140S, Q148H	
	Homemade NGS-HIVdb	T97A (100%), E138K (99%), G140S (99%), Q148H (99%)	
	AD4SEQ-SmartVir	T97A (94%), E138K (94%), G140S (93%), Q148H (94%)	
**6003**	Sanger-HIVdb	R263K	
	Homemade NGS-HIVdb	R263K (96%)	
	AD4SEQ-SmartVir	R263K (98%)	
**6669**	Sanger-HIVdb	None	
	Homemade NGS-HIVdb	**E157Q (7.1%)**	
	AD4SEQ-SmartVir	**E157Q (11%)**	
**6813**	Sanger-HIVdb	L74M, G140S, Q148K	
	Homemade NGS-HIVdb	L74M (98%) G140S (99%), Q148K (98%)	
	AD4SEQ-SmartVir	L74M (97%), G140S (97%), Q148K (98%)	
**6835**	Sanger-HIVdb	None	
	Homemade NGS-HIVdb	None	
	AD4SEQ-SmartVir	**L74M (10%)**	
**6880**	Sanger-HIVdb	Q95QK	
	Homemade NGS-HIVdb	Q95K (35%)	
	AD4SEQ-SmartVir	Q95K (36%)	
**7347**	Sanger-HIVdb	None	
	Homemade NGS-HIVdb	**T97A (6%)**	
	AD4SEQ-SmartVir	None	

**Table 4 viruses-16-01465-t004:** Regions with a coverage depth of <100 reads per base according to the NGS system and the data processing method. There were no low-coverage regions within the protease coding regions.

Sample	NGS System and Data Processing Method	Region and Relative Amino Acids with Coverage < 100×
5826	AD4SEQ-SmartVir	RT 14–49
	AD4SEQ-HIVdb	RT 14–49
	Homemade NGS-HIVdb	IN 1–25
5974	AD4SEQ-SmartVir	IN 201–284
	AD4SEQ-HIVdb	IN 201–288
	Homemade NGS-HIVdb	None
5979	AD4SEQ-SmartVir	RT 14–49, 223–235/IN 1–75, 201–284
	AD4SEQ-HIVdb	RT 14–49, 223–235/IN 1–75, 201–288
	Homemade NGS-HIVdb	None
6003	AD4SEQ-SmartVir	RT 260–275
	AD4SEQ-HIVdb	RT 260–319
	Homemade NGS-HIVdb	None
6084	AD4SEQ-SmartVir	None
	AD4SEQ-HIVdb	IN 201–209
	Homemade NGS-HIVdb	None
6107	AD4SEQ-SmartVir	IN 1–10, 66–75, 159–171
	AD4SEQ-HIVdb	IN 1–11, 66–75
	Homemade NGS-HIVdb	None
6216	AD4SEQ-SmartVir	RT 260–319
	AD4SEQ-HIVdb	RT 260–319
	Homemade NGS-HIVdb	IN 1–39
6222	AD4SEQ-SmartVir	IN 1–75
	AD4SEQ-HIVdb	IN 1–75
	Homemade NGS-HIVdb	None
6363	AD4SEQ-SmartVir	RT 260–275; IN 201–284
	AD4SEQ-HIVdb	IN 201–288
	Homemade NGS-HIVdb	None
6408	AD4SEQ-SmartVir	None
	AD4SEQ-HIVdb	IN 201–288
	Homemade NGS-HIVdb	None
6436	AD4SEQ-SmartVir	None
	AD4SEQ-HIVdb	RT 260–319, 359
	Homemade NGS-HIVdb	None
6471	AD4SEQ-SmartVir	RT 260–319
	AD4SEQ-HIVdb	RT 260–319
	Homemade NGS-HIVdb	None
6493	AD4SEQ-SmartVir	RT 260–319
	AD4SEQ-HIVdb	RT 260–319
	Homemade NGS-HIVdb	None
6592	AD4SEQ-SmartVir	IN 1–31, 66–75
	AD4SEQ-HIVdb	IN 1–75
	Homemade NGS-HIVdb	None
6880	AD4SEQ-SmartVir	IN 74–75
	AD4SEQ-HIVdb	IN 73–75
	Homemade NGS-HIVdb	None
7312	AD4SEQ-SmartVir	RT 14–49
	AD4SEQ-HIVdb	RT 14–49
	Homemade NGS-HIVdb	None
7347	AD4SEQ-SmartVir	None
	AD4SEQ-HIVdb	RT 35–49
	Homemade NGS-HIVdb	None

RT, reverse transcriptase; IN, integrase.

**Table 5 viruses-16-01465-t005:** Discordant identification of drug resistance mutations between SmartVir and HIVdb interpretation tools from FASTQ files generated by the AD4SEQ sequencing method. Mutations in bold are those identified by only one interpretation tool. The frequency of each mutation is shown in brackets. There were no discordant cases with integrase inhibitor resistance mutations.

Sample	Data Processing Method	PIs	NRTIs	NNRTIs
5974	SmartVir	None	None	K103N (99%), N348I (99%)
	HIVdb	None	**S68G (98%)**	K103N (99%), N348I (99%)
6003	SmartVir	None	M184V (98%)	K103N (95%), K238N (96%)
	HIVdb	None	**S68G (91%)**, M184V (98%)	K103N (93%), K238N (96%)
6006	SmartVir	None	None	K103N (99%), N348I (99%)
	HIVdb	None	**S68G (98%)**	K103N (99%), N348I (99%)
6436	SmartVir	None	None	K103N (99%)
	HIVdb	None	**S68G (81%)**	K103N (99%)
6471	SmartVir	None	M41L (98%), D67N (19%), M184V (91%), L210W (98%), T215Y (92%), K219E (9%)	L100I (99%), K103N (97%), N348I (99%)
	HIVdb	None	M41L (99%), **E44A (15%)**, D67N (19%), M184V (92%), L210W (97%), T215Y (95%), K219E (11%)	L100I (99%), K103N (97%), N348I (100%)
6493	SmartVir	None	D67G (98%), K70R (99%), M184V (92%), T215I (94%), K219E (87%)	K103N (98%), V108I (99%), Y181C (26%), K238T (97%), N348I (99%)
	HIVdb	None	D67G (97%), **S68G (98%)**, K70R (99%), M184V (93%), T215I (96%), K219E (89%)	K103N (99%), V108I (98%), Y181C (25%), K238T (94%), N348I (100%)
6817	SmartVir	None	None	K103N (69%)
	HIVdb	None	None	K103N (69%), **V179T (94%)**
6835	SmartVir	None	M41L (19%), L210W (98%), T215DS (D: 25%, S: 72%)	None
	HIVdb	**M46I (5%)**	M41L (19%), L210W (99%), T215DS (D: 25%, S: 73%)	None
6880	SmartVir	None	None	E138G (97%)
	HIVdb	None	None	E138G (96%), **V179T (6%)**
7312	SmartVir	None	**D67NE (N: 8%, E: 89%)**, T69NG (N: 6%, G: 89%), K219Q (96%)	A98G (97%), V106I (98%), Y181C (66%)
	HIVdb	None	**D67ΔN (Δ: 8%, N: 51%), S68G (8%)**, T69NG (N: 6%, G:88%), K219Q (98%)	A98G (97%), V106I (98%), Y181C (68%)

PIs, protease inhibitors; NRTIs, nucleoside reverse transcriptase inhibitors; NNRTI, non-NRTI.

**Table 6 viruses-16-01465-t006:** Discordant cases of predicted drug susceptibility according to the sequencing system and to the data processing method.

Sample	Sequencing System and Data Processing Method	PIs			NRTIs			NNRTIs					INIs		
		ATV	LPV	DRV	ABC	TAF	3TC/FTC	DOR	RPV	ETR	EFV	NVP	DTG	BIC	CAB
**5979**	Sanger-HIVdb	R	I	LLR	I	I	S	PLLR	R	R	R	R	R	R	R
	Homemade NGS-HIVdb	R	I	LLR	I	I	S	PLLR	R	R	R	R	R	R	R
	AD4SEQ-SmartVir	R	I	LLR	LLR	LLR	S	PLLR	R	R	R	R	R	R	R
	AD4SEQ-HIVdb	R	I	LLR	LLR	LLR	S	PLLR	R	R	R	R	R	R	R
**6471**	Sanger Sequencing	S	S	S	R	R	R	I	R	I	R	R	S	S	S
	Homemade NGS	S	S	S	R	I	R	I	R	I	R	R	S	S	S
	AD4SEQ-SmartVir	S	S	S	R	R	R	I	R	I	R	R	S	S	S
	AD4SEQ-HIVdb	S	S	S	R	R	R	I	R	I	R	R	S	S	S
**6493**	Sanger Sequencing	S	S	S	I	LLR	R	PLLR	S	S	R	R	S	S	S
	Homemade NGS	S	S	S	I	LLR	R	I	I	I	R	R	S	S	S
	AD4SEQ-SmartVir	S	S	S	I	LLR	R	I	I	I	R	R	S	S	S
	AD4SEQ-HIVdb	S	S	S	I	LLR	R	I	I	I	R	R	S	S	S
**6592**	Sanger Sequencing	S	S	S	S	S	S	S	I	PLLR	PLLR	PLLR	S	S	S
	Homemade NGS	S	S	S	S	S	S	LLR	R	LLR	LLR	I	S	S	S
	AD4SEQ-SmartVir	S	S	S	S	S	S	LLR	R	LLR	LLR	I	S	S	S
	AD4SEQ-HIVdb	S	S	S	S	S	S	LLR	R	LLR	LLR	I	S	S	S
**6669**	Sanger Sequencing	S	S	S	S	S	S	I	S	S	R	R	S	S	S
	Homemade NGS	S	S	S	S	S	S	I	S	S	R	R	S	S	S
	AD4SEQ-SmartVir	S	S	S	S	S	S	I	I	I	R	R	S	S	S
	AD4SEQ-HIVdb	S	S	S	S	S	S	I	I	I	R	R	S	S	S
**6835**	Sanger Sequencing	S	S	S	S	S	S	S	S	S	S	S	S	S	S
	Homemade NGS	S	S	S	LLR	LLR	S	S	S	S	S	S	S	S	S
	AD4SEQ-SmartVir	S	S	S	LLR	LLR	S	S	S	S	S	S	S	S	PLLR
	AD4SEQ-HIVdb	PLLR	PLLR	S	LLR	LLR	S	S	S	S	S	S	S	S	PLLR
**7312**	Sanger Sequencing	S	S	S	I	I	LLR	LLR	LLR	LLR	LLR	I	S	S	S
	Homemade NGS	S	S	S	I	I	I	I	R	I	I	R	S	S	S
	AD4SEQ-SmartVir	S	S	S	LLR	LLR	S	I	R	I	I	R	S	S	S
	AD4SEQ-HIVdb	S	S	S	I	I	LLR	I	R	I	I	R	S	S	S
**7347**	Sanger Sequencing	S	S	S	S	S	S	LLR	R	I	I	R	S	S	S
	Homemade NGS	S	S	S	S	S	S	I	R	I	I	R	S	S	S
	AD4SEQ-SmartVir	S	S	S	S	S	S	I	R	I	I	R	S	S	S
	AD4SEQ-HIVdb	S	S	S	S	S	S	I	R	I	I	R	S	S	S

PIs, protease inhibitors; NRTIs, nucleoside reverse transcriptase inhibitors; NNRTI, non-NRTI; INIs, integrase inhibitors; ATV, atazanavir; LPV, lopinavir; DRV, darunavir; ABC, abacavir; TAF, tenofovir alafenamide; 3TC, lamivudine; FTC, emtricitabine; DOR, doravirine; RPV, rilpivirine; ETR, etravirine; EFV, efavirenz; NVP, nevirapine; DTG, dolutegravir; BIC, bictegravir; CAB, cabotegravir; S, full susceptibility (green); PLLR, potential low-level resistance (yellow); LLR, low-level resistance (light orange); I, intermediate resistance (orange); R, high-level resistance (red).

## Data Availability

Data are contained within this article and Appendix A. All data not included are available on request from the corresponding author.

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
