# Peer review of "A Comparison of Sanger Sequencing and Amplicon-Based Next Generation Sequencing Approaches for the Detection of HIV-1 Drug Resistance Mutations"

_viruses, 2024, doi:10.3390/v16091465_

Round 1

Reviewer 1 Report

Comments and Suggestions for Authors

In the present manuscript, Biba and colleagues aimed to compare the sensitivity of two NGS methods in identifying resistance associated mutations detected by Sanger sequencing in clinical HIV-1 samples: a homemade NGS amplicon-based protocol and the AD4SEQ HIV-1 Solution v2 12 (AD4SEQ) NGS kit from Arrow Diagnostics. Even though the study is limited on a relatively small number of samples (n=28), this is the first paper analyzing the performance of AD4SEQ system. The results obtained provide useful information on the performance and caveats of AD4SEQ system in the clinical setting. This is relevant by considering the fact most laboratories are transitioning from Sanger to NGS for HIV-1 genotypic drug resistance testing. Beyond the small size, other limitations of the work are the limited representation of HIV-1 subtypes and the almost total absence of samples at viremia levels below 1000 copies/mL. On the other hand, authors are conscious of these aspects, as reported in the discussion section. The manuscript is well written and clear. I have only two small comments, as reported below.

1) If I well understand, the minimum read depth parameter was set at >100 reads per each position with a mutation detection threshold of 5%, regardless the fact that this position was associated with resistance. Is this correct?

2) Table 3: I would prefer that in case of absence of detection of mutation authors report in the table “None” instead of the “dash”.

Author Response

Reply to reviewer 1

In the present manuscript, Biba and colleagues aimed to compare the sensitivity of two NGS methods in identifying resistance associated mutations detected by Sanger sequencing in clinical HIV-1 samples: a homemade NGS amplicon-based protocol and the AD4SEQ HIV-1 Solution v2 12 (AD4SEQ) NGS kit from Arrow Diagnostics. Even though the study is limited on a relatively small number of samples (n=28), this is the first paper analyzing the performance of AD4SEQ system. The results obtained provide useful information on the performance and caveats of AD4SEQ system in the clinical setting. This is relevant by considering the fact most laboratories are transitioning from Sanger to NGS for HIV-1 genotypic drug resistance testing. Beyond the small size, other limitations of the work are the limited representation of HIV-1 subtypes and the almost total absence of samples at viremia levels below 1000 copies/mL. On the other hand, authors are conscious of these aspects, as reported in the discussion section. The manuscript is well written and clear. I have only two small comments, as reported below.

Reply to Reviewer 1. Thank you very much for appreciating our work.

Point 1. If I well understand, the minimum read depth parameter was set at >100 reads per each position with a mutation detection threshold of 5%, regardless the fact that this position was associated with resistance. Is this correct?

Reply to point 1, reviewer 1: Yes, we analyzed the samples with a minimum read depth of 100 reads per position and a mutation detection threshold of 5% for both SmartVir and HIVdb. This was applied to any position independently from being associated with drug resistance or not. We have clarified this in the manuscript.

Point 2. Table 3: I would prefer that in case of absence of detection of mutation authors report in the table “None” instead of the “dash”.

Reply to point 2, reviewer 1: Thank you for your observation. The table was modified as suggested and the same was applied to the other tables where appropriate.

Reviewer 2 Report

Comments and Suggestions for Authors

The manuscript: "Comparison of Sanger sequencing and amplicon-based NGS approaches for the detection of HIV-1 drug resistance mutations" does not have a large study group, but it is still sufficient to asses the differences between two methods. The introduction contains enough background for the subject in question and the methods are thoroughly described. The result part however needs some clarity. The tables are too robust and some dividing would help. The entire Table 3. would be better suited for the supplement and the one in the article should contain only samples that have differences detected when NGS method is applied. Discussion offers enough studies by other authors for comparison with the results obtained in this study and the conclusion states some of the limitation of this study. Higher number of tested samples, perhaps some without any mutations detected using Sanger sequencing would be a valuable addition to this paper, but I understand that limited resources prevent it. It would be useful to state whether there were any discrepancies when it comes to detecting the HIV subtype using different sequencing methods, if I missed it somewhere in the text I apologize. Other than that, I find the manuscript suitable for publishing.

Comments on the Quality of English Language

The English language needs only minor interventions, easily resolved during the editing process before publishing. 

Author Response

Reply to Reviewer 2

The manuscript: "Comparison of Sanger sequencing and amplicon-based NGS approaches for the detection of HIV-1 drug resistance mutations" does not have a large study group, but it is still sufficient to assess the differences between two methods. The introduction contains enough background for the subject in question and the methods are thoroughly described. The result part however needs some clarity. The tables are too robust and some dividing would help. The entire Table 3. would be better suited for the supplement and the one in the article should contain only samples that have differences detected when NGS method is applied. Discussion offers enough studies by other authors for comparison with the results obtained in this study and the conclusion states some of the limitation of this study. Higher number of tested samples, perhaps some without any mutations detected using Sanger sequencing would be a valuable addition to this paper, but I understand that limited resources prevent it. It would be useful to state whether there were any discrepancies when it comes to detecting the HIV subtype using different sequencing methods, if I missed it somewhere in the text I apologize. Other than that, I find the manuscript suitable for publishing.

Reply to point 1: Thank you very much for appreciating our work. We agree that Table 3 is quite large, however we feel that the information shown in Table 3 are all relevant for the purpose of our work and we would like to avoid moving any part to the supplementary material. To make the data more readable, we have split the table into three smaller tables, one for each target, PR, RT, IN and we moved the original table to the Supplementary data section. We believe this solution meets the request made by the reviewer and saves completeness and clarity of data.

The limited representativeness of HIV subtypes in the data set did not allow to compare subtype B with other individual subtypes. As indicated in the Results section, we did not observe a larger rate of low-coverage cases with non-B subtypes for AD4SEQ-SmartVir, which was the system most affected by low coverage issues. However, a recent paper published by Armenia et al. (https://doi.org/10.3390/v16091422) detected non-B subtype and low-level viremia as factors associated with low coverage in a wide study comparing HIV drug resistance Interpretation tools for NGS data. We have added this reference in the Discussion section.